# From Cross-Sectional CT to Dynamic Insights: Pseudotime-Based Modeling of Lung Nodule Progression

**Luoting Zhuang**[1] [iD]                                      LUOTINGZHUANG@G.UCLA.EDU
**Linh M. Tran**[2,3] [iD]                                      LINHMTRAN@MEDNET.UCLA.EDU
**Yunzheng Zhu**[1] [iD]                                        YUNZHENGZHU19@G.UCLA.EDU
**Ashley E. Prosper**[1] [iD]                                   APROSPER@MEDNET.UCLA.EDU
**William Hsu**[1] [iD]                                         WHSU@MEDNET.UCLA.EDU

[1] *Medical & Imaging Informatics, Department of Radiological Sciences, David Geffen School of Medicine at UCLA, Los Angeles, CA, USA*
[2] *Department of Medicine, Division of Pulmonology and Critical Care, David Geffen School of Medicine at UCLA, Los Angeles, CA, USA*
[3] *VA Greater Los Angeles Healthcare System, Los Angeles, CA, USA*

**Editors:** Accepted for publication at MIDL 2026

## Abstract

Early detection of lung cancer relies on a comprehensive understanding of the progression of pulmonary nodules. Existing longitudinal modeling approaches are constrained due to the limited availability of longitudinal datasets and the failure to capture the inter-nodular relationship. In this study, we present one of the first applications of pseudotime inference, adapted from single-cell RNA sequencing studies, to reconstruct progression trajectories of nodules from cross-sectional CT images. We collected 13,626 nodule snapshots from two screening cohorts and reserved a longitudinal test set for evaluation. We compared a graph-based pseudotime method, diffusion pseudotime, and an unsupervised deep learning framework combining a variational autoencoder and a neural ordinary differential equation. Both approaches demonstrate longitudinal consistency, with malignant nodules showing a higher correlation between pseudotime and actual time. Pseudotime aligns with clinically relevant features such as irregular margins and solid consistency. Furthermore, pseudotime and delta-pseudotime effectively stratify nodules into distinct malignancy risk groups and remain significant independent predictors of malignancy after adjusting for established semantic biomarkers. Our study highlights pseudotime inference as a promising tool for dynamic modeling of lesion progression using static imaging data. The implementation code is available at https://github.com/luotingzhuang/Pseudotime4Nodules.

**Keywords:** Pseudotime inference, disease trajectory, lung nodules, diffusion maps, unsupervised learning, medical imaging biomarkers

## 1. Introduction

Lung cancer remains the leading cause of cancer-related mortality, accounting for approximately 350 deaths per day in the United States (Siegel et al., 2023). Research studies suggest that implementing screening programs with low-dose computed tomography (LDCT) facilitates early lung cancer detection and results in a significant reduction in lung cancer mortality (Team, 2011; Potter et al., 2022). Understanding the progression of pulmonary

nodules is essential for the accurate detection of lung cancer. Multiple prior works have utilized convolutional neural networks, recurrent neural networks, and transformer models to predict lung cancer risk from longitudinal CT scans (Ardila et al., 2019; Gao et al., 2019; Li et al., 2023). However, these approaches have two limitations. First, these methods primarily focus on the local evolution of individual nodules and ignore the general progression patterns shared across different nodules. Therefore, the learned latent space can lack global structure, making it harder to capture clinically relevant dynamic characteristics across nodules. Second, longitudinal data are typically small, which limits model training across diverse samples and increases the risk of overfitting. For instance, the National Lung Screening Trial (NLST) is the largest publicly available longitudinal dataset for lung cancer screening, serving as a primary resource for nearly all longitudinal modeling studies. Meanwhile, large collections of single-timepoint "static" CT scans have been underutilized for learning nodule evolution.

Unlike conventional imaging analyses that treat nodules as isolated observations, pseudotime inference in single-cell RNA sequencing (scRNA-seq) studies has been widely used to infer cell differentiation or progression trajectories from static snapshots (Trapnell et al., 2014; Street et al., 2018; Haghverdi et al., 2016). The underlying assumption is that, although temporal information is lost when cells are sampled from tissue, they still reside along a latent developmental trajectory. By leveraging gene expression profiles, pseudotime algorithms reconstruct dynamic processes from these snapshots, ordering cells along a continuous trajectory that represents biological processes. Traditional algorithms typically follow two steps: (1) meaningful representations and relationships between cells are derived using techniques such as manifold learning, clustering, or graph-based methods, and (2) a root cell is selected, and pseudo-temporal ordering (pseudotime) is assigned to each cell using approaches such as shortest-path algorithms or diffusion maps (Cannoodt et al., 2016). More recently, research has been increasingly focused on leveraging deep learning (DL) models to learn complex, nonlinear relationships, generate batch-normalized, robust embeddings, and automatically infer pseudotime (Lopez et al., 2018; Li, 2023).

There are conceptual parallels between scRNA-seq data and imaging features extracted from pulmonary nodules. Similar to cells, nodules from different individuals can also reflect stages along a continuum of nodule development, from benign to malignant transformation. Imaging features extracted from CT scans capture the dynamic nature of pulmonary nodules, which undergo gradual changes over time in terms of size, shape, texture, intensity, and vascularity. This perspective enables us to apply trajectory inference methods to imaging features, allowing for the reconstruction of developmental or pathological pathways. Several studies have explored using pseudotime inference techniques to model the progression of Alzheimer's disease (He et al., 2024; Glazman et al., 2025), rectal cancer (Lee et al., 2024), and lung adenocarcinoma (Qiu et al., 2025) using imaging features.

We hypothesize that applying pseudotime inference to imaging features from cross-sectional data can construct a trajectory of nodule progression stages that captures the collective evolution of nodules. In this context, we introduce pseudotime (biological state) and delta-pseudotime (change in states over time) as biologically informed biomarkers to assist clinicians in inferring nodule malignancy. This study is among the first to apply pseudotime inference to medical imaging for modeling the progression of lesions. Further-

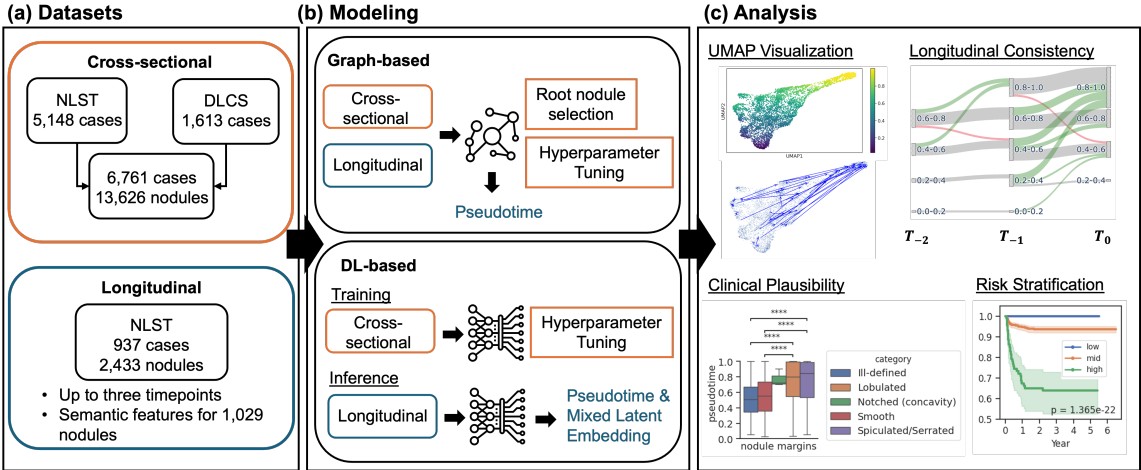

Figure 1: **Pipeline Overview.** **(a)** We collected cross-sectional data from the National Lung Screening Trial (NLST) and the Duke Lung Cancer Screening (DLCS) for modeling. A subset of longitudinal data from the NLST was used for validation. **(b)** In graph-based methods, cross-sectional and longitudinal data were integrated in constructing graphs. However, hyperparameter tuning and root nodule selection were performed based on cross-sectional data only. In contrast, the deep learning (DL)-based method used cross-sectional data for training and hyperparameter optimization, while longitudinal data were used during inference to estimate pseudotime and latent embeddings. **(c)** We evaluated the validity and utility of pseudotime through longitudinal consistency, clinical plausibility, risk stratification, and Cox proportional hazards modeling.

more, we present a comparative analysis of two approaches, a traditional graph-based and a DL-based method.

## 2. Methods

### 2.1. Data Preprocessing

We used data from the NLST, which includes LDCT scans collected at up to three annual screening timepoints. We filtered out 6,995 cases with an indeterminate pulmonary nodule at least once during screening. We applied two pre-trained nodule detection models, MONAI (Cardoso et al., 2022) and Liao et al. (Liao et al., 2019). To reduce false positives, we matched the z-axis of the nodule detected by the algorithms to the slice number recorded in the Abnormality table provided by the NLST organizers. Additionally, we implemented the MaskedSeg model (Zhuang et al., 2025) to segment the lung and remove nodules that fall outside the lung region. In addition to the algorithm-detected nodules, we further incorporated three datasets with manually annotated nodules in NLST, including an in-house dataset from UCLA, LUNA25 (Peeters et al., 2025), and NLST annotations released by the developers of the Sybil lung cancer risk model (Mikhael et al., 2023). We obtained a total of 6,085 cases (19,588 nodules), of which 5,635 have longitudinal CT scans. We tracked

the detected nodules across timepoints using a registration algorithm that combines affine and correspondence field registration (Heinrich et al., 2015), and identified 12,604 unique nodules across 5,636 cases. In total, we have 6,085 cases with 13,514 nodules.

In Figure 1, we summarize the dataset used for modeling and evaluation. We selected 937 cases with 2,433 nodules from the NLST to create a longitudinal cohort for the held-out evaluation set. Among these cases, semantic features such as nodule consistency, shape, and size had been annotated for 1,029 nodules. Since our study examines whether cross-sectional data can help infer nodule trajectory, we included only the nodule at the final timepoint to form a cross-sectional cohort with 5,148 cases. Additionally, we collected LDCT scans from 1,613 individuals in the Duke Lung Cancer Screening (DLCS) dataset (Wang et al., 2025), which only contains cross-sectional scans. Combining the cross-sectional NLST and DLCS cohorts, we obtained 6,761 cases with 13,626 nodules for training.

## 2.2. Feature Extraction

We extracted imaging features from each nodule, similar to how gene expression profiles are derived in scRNA-seq analysis. We opted to extract imaging features using a lesion foundation model called the Foundation Model for Cancer Imaging Biomarkers (FMCIB) (Pai et al., 2024). FMCIB was trained with contrastive learning to distinguish volumes with and without lesions. FMCIB features have shown to capture morphological characteristics and demonstrate strong associations with biological markers of malignancy. The CT scans were resampled to a voxel spacing of $1 \times 1 \times 1$mm, and boxes measuring $50 \times 50 \times 50$ voxels were cropped around the nodule's centroid. We fed each nodule crop into the model to generate a 4,096-dimensional deep feature embedding.

## 2.3. Experimental Setup

### 2.3.1. PSEUDOTIME INFERENCE

We investigated two pseudotime analysis methods used for scRNA-seq data: a traditional graph-based method and a DL-based method.

**Graph-Based Trajectory Inference.** In Figure 2a, we illustrate the pipeline of leveraging a graph-based pseudotime method with diffusion pseudotime (Haghverdi et al., 2016). First, FMCIB features were used to construct a k-nearest neighbor (KNN) graph. We utilized the Leiden community detection algorithm (Traag et al., 2019) on the graph to obtain clusters of nodules. In most cases, the root nodule was randomly selected from the cluster with the lowest malignancy percentage. However, we observed that trajectories often originate from multiple starting points but converge toward a single destination. To better capture this progression, we randomly selected the root nodule from the cluster with the highest malignancy percentage in the cross-sectional data. Pseudotime was then computed starting from this root; the final pseudotime score was defined as $1 -$ pseudotime.

Diffusion pseudotime first computed diffusion maps, which relied on the Markov transition matrix $P$ to model random walks on the graph. Nodules that were similar to each other exhibited similar transition probabilities of reaching other cells in the graph. Diffusion maps were particularly effective at capturing the global structure of high-dimensional data and were robust to noise. The pseudotime of nodule $i$ was then derived by the Euclidean distance from the root nodule $r$ within the diffusion space. Specifically, let $P^t$ represent the

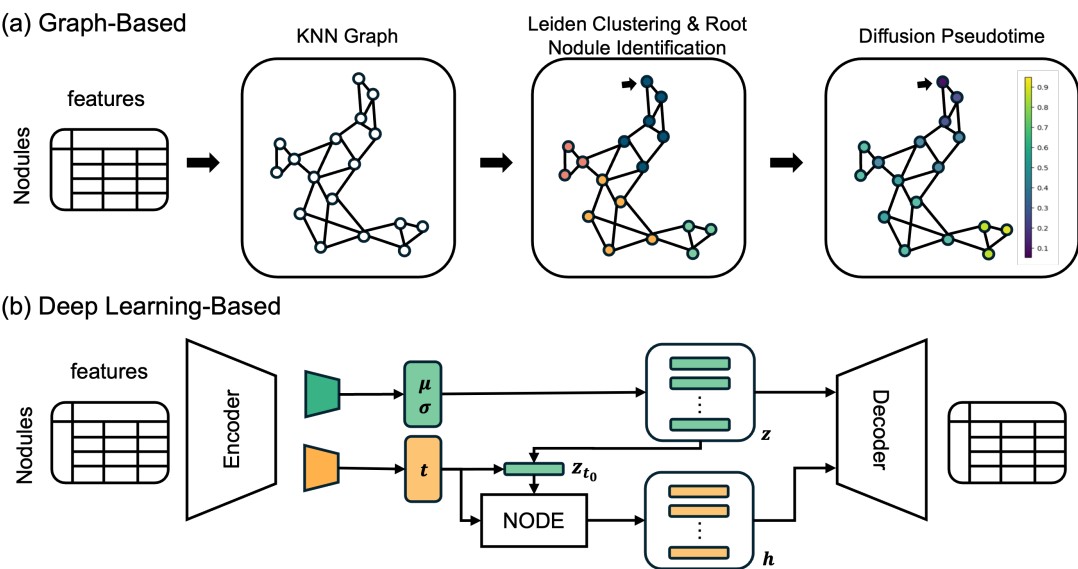

Figure 2: **Pseudotime Inference Algorithms.** **(a)** Graph-based method. A k-nearest neighbor (KNN) graph was constructed from imaging features. The Leiden algorithm was applied to identify communities within the graph. After selecting a root nodule, diffusion pseudotime was computed to assign a temporal ordering to all nodules. **(b)** Deep learning-based method. A variational autoencoder architecture was employed. In the green subsection, the encoder estimated the mean ($\mu$) and standard deviation ($\sigma$) of a Gaussian distribution, from which latent embeddings ($z$) were sampled. In the orange branch, the encoder predicted pseudotime ($t$) for each nodule. Based on the smallest pseudotime value, the corresponding embedding $z_{t_0}$ was selected as the initial state. A neural ordinary differential equation (NODE) model then integrated $z_{t_0}$ and $t$ to produce a transformed latent representation ($h$). Both $z$ and $h$ were passed to the decoder for feature reconstruction.

t-step transition matrix and $\phi_0(k)$ the stationary distribution. The pseudotime for nodule $i$, the diffusion distance between nodule $i$ and root nodule $r$ is defined as:

$$\text{Pseudotime}_i = D_t^2(r, i) = \sum_{k=1}^n \frac{\left[(P^t)_{rk} - (P^t)_{ik}\right]^2}{\phi_0(k)}$$

**DL-Based Trajectory Inference.** We implemented the scTour framework (Li, 2023) to infer pseudotime trajectories in our dataset (Figure 2b). scTour is a DL-based pseudotime analysis algorithm that uses self-supervised learning, combining a variational autoencoder (VAE) and neural ordinary differential equations (NODE). In VAE, the encoder mapped FMCIB features into two vector outputs, the mean ($\mu$) and the standard deviation ($\sigma$) of a Gaussian latent distribution. The latent embedding ($z$) was randomly sampled from the distribution and fed into the decoder to reconstruct the nodule's deep features. At the same time, an additional linear layer was attached to the encoder to predict pseudotime ($t$) for each nodule. Starting from the latent embedding of the nodule with the smallest pseudotime ($z_{t_0}$) and the set of predicted timepoints, the NODE module generated a continuous

trajectory of latent states $(h_t)$. Each latent representation was then passed through the same decoder to reconstruct the features. This design enforced that the latent dynamics inferred at each timepoint remain consistent with the observed data, thereby enabling the model to learn a meaningful pseudotime ordering. The latent embeddings $z$ and $h$ were averaged with equal weights to form a mixed representation for downstream analysis.

The model was optimized with four loss functions. First, two reconstruction losses were computed from latent embedding $z$ and $h$, using mean squared error (MSE) to reconstruct the original imaging feature $x$. Second, we minimized Kullback–Leibler (KL) divergence to prompt a well-structured latent space that follows a Gaussian prior. Finally, a MSE loss was used to enforce consistency between latent embeddings $z$ and $h$. We set the scaling factors for reconstruction losses ($\alpha_z$ and $\alpha_h$) to be 0.5, and those for the KL divergence ($\beta$) and the consistency loss ($\gamma$) to be 1.

$$\mathcal{L} = \underbrace{-\alpha_z \cdot \log p(x \mid z) - \alpha_h \cdot \log p(x \mid h)}_{\text{Reconstruction Loss on z and h}} + \underbrace{\beta \cdot D_{\text{KL}}\big(q(z \mid x) \,\|\, p(z)\big)}_{\text{VAE KL divergence}} - \underbrace{\gamma \cdot \log p(z \mid h)}_{\text{Consistency Loss}}$$

### 2.3.2. Implementation and Reproducibility

The diffusion pseudotime was implemented using a single-cell analysis Python package, Scanpy v1.11.5 (Wolf et al., 2018). The DL-based method was implemented based on the scTour framework on GitHub[1]. The model was trained on an NVIDIA Quadro RTX 8000 GPU with 48 GB of memory. We released our implementation code on GitHub[2].

Hyperparameters for diffusion pseudotime were tuned via grid search to maximize the alignment between inferred pseudotime and malignancy label in the cross-sectional cohort. The search space included the number of neighbors in KNN (30,50,100,150,200,300) and the number of diffusion map components (5,10,15) used to compute pseudotime. Spearman correlation was used to assess the alignment. The optimal configuration with the highest correlation was 50 nearest neighbors and 10 diffusion map components. Similarly, for the VAE-NODE model, we performed hyperparameter tuning for latent embedding dimensions (16,64,128), batch sizes (256,512,4096), and learning rates (1e-3, 5e-4). Based on the highest Spearman correlation, we selected the model trained with a latent embedding dimension of 128, a batch size of 512, and a learning rate of 5e-4. The model was trained for 400 epochs.

### 2.4. Evaluation

For each of the nodules in the longitudinal NLST set, we extracted the pseudotime and the delta-pseudotime, which we defined as the change in pseudotime from the prior timepoint. To assess the validity and utility of pseudotime, we conducted four evaluations. (1) **Temporal validity**: We visualized FMCIB feature embeddings in Uniform Manifold Approximation and Projection (UMAP) for the graph-based method (Figure 3). The mixed latent embeddings were used for computing UMAP for the DL-based method. We overlaid pseudotime values and the trajectories of malignant and benign nodules on UMAP, marked with arrows, to assess the concordance between inferred pseudotime and observed

---

1. https://github.com/LiQian-XC/sctour
2. https://github.com/luotingzhuang/Pseudotime4Nodules

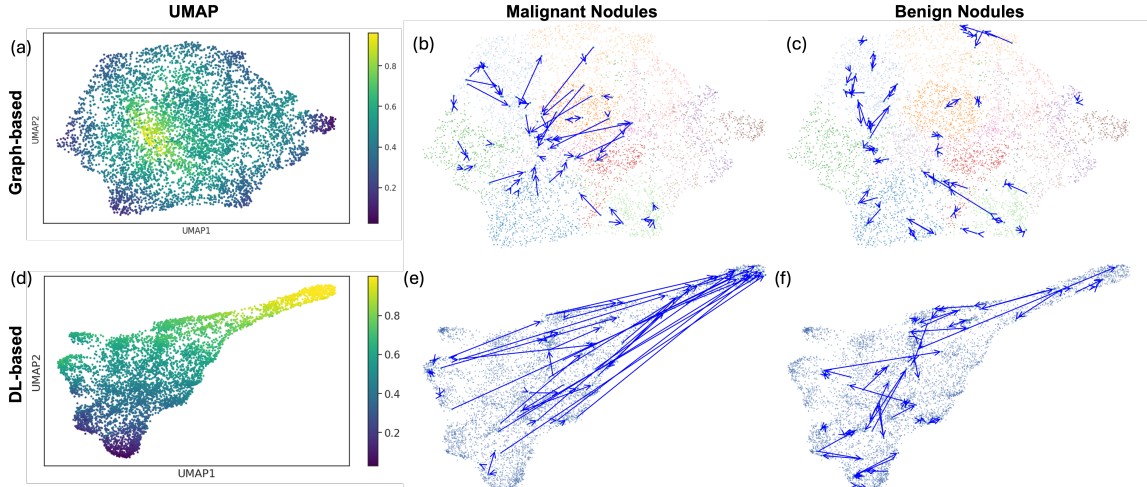

Figure 3: **UMAP Visualizations.** Pseudotime is colored on UMAP on subplots **a and b**. In the graph-based method, the UMAP was constructed using FMCIB features, whereas in the deep learning-based method, it was derived from a mixed latent embedding. Trajectories of malignant **(b, e)** and benign nodules **(c, f)** are illustrated. Each arrow in the graph represents the change observed in a single nodule over approximately a one-year period.

longitudinal evolution. In addition, we computed the Pearson correlation between pseudotime and actual time for benign and malignant nodules (Figure 4). The Mann–Whitney U test was performed to evaluate the statistical significance of the Pearson correlation. To visualize the transition dynamics of pseudotime across chronological timepoints ($T_{-2}$ to $T_0$), Sankey diagrams were constructed with pseudotime values discretized into bins of 0.2. $T_0$ denotes the last imaging timepoint and also corresponds to the time of lung cancer diagnosis in malignant cases. Within these diagrams, the thickness of each flow represents the relative number of nodules. Increasing pseudotime transitions were colored in green, decreasing pseudotime in red, and stable pseudotime in gray. (2) **Clinical Plausibility**: We compared pseudotime across nodules with different semantic features, including nodule margins, shape, consistency, and longest axial diameter (Figure 5). Hypothesis testing was performed using the Mann-Whitney U test at the 0.05 significance level. We corrected multiple testing using Benjamini-Hochberg. (3) **Risk Stratification**: We stratified nodules into three groups (low, medium, and high) with equal width in pseudotime and delta pseudotime, separately. We plotted Kaplan-Meier curves to show the cumulative probability of remaining benign over time for each group (Figure 6). The log-rank test was performed to evaluate whether there is a statistically significant difference in the time-to-diagnosis distributions between groups. (4) **Independent Predictive Value**: We fitted a Cox proportional hazards model on pseudotime and delta-pseudotime to analyze time to lung cancer diagnosis (Table 1). This model was adjusted for semantic features (size, change in size, margin, shape, consistency, and change in consistency), which are known to be lung cancer biomarkers. We reported the hazard ratio (HR), its 95% confidence interval, and the p-values for each variable.

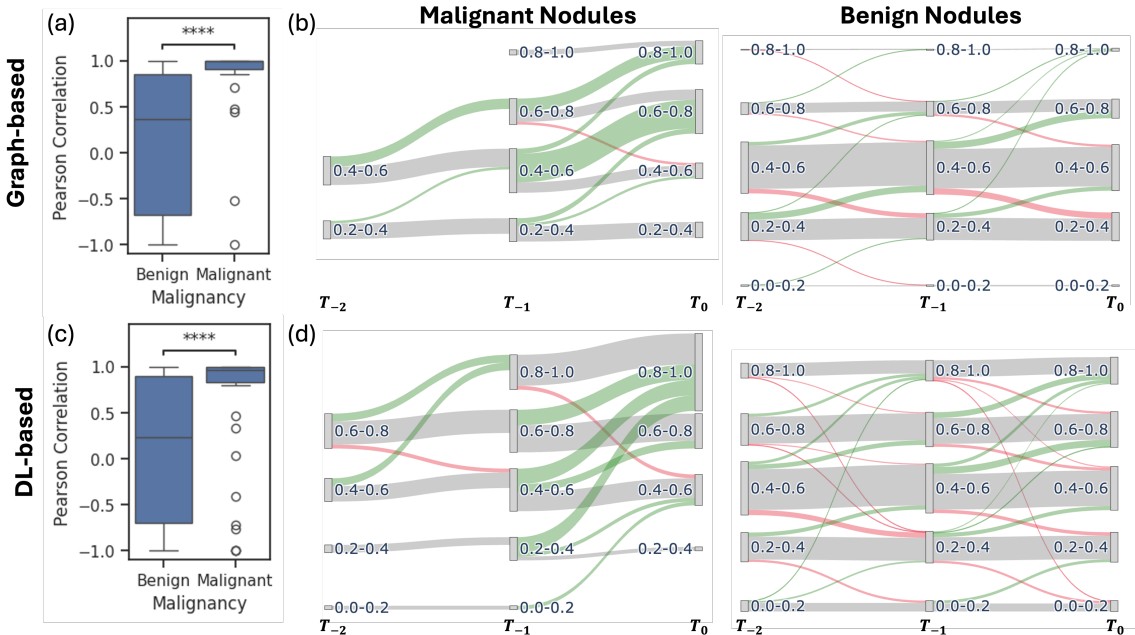

Figure 4: **Longitudinal Consistency.** Boxplots **(a, c)** present Pearson correlations between inferred pseudotime and actual time for benign and malignant nodules. Statistical significance was indicated by asterisks (*) based on the Mann–Whitney U test. Sankey diagrams **(b, d)** depict pseudotime transitions of longitudinal scans. $T_0$ denotes the final scan timepoint. Pseudotime is binned into 0.2-interval bins. Green flow indicates transitions toward higher pseudotime, red indicates a decrease in pseudotime, and gray indicates stable pseudotime. The thickness of flows represents the relative number of nodules.

## 3. Results

### 3.1. Increasing pseudotime over time in malignant nodules validates its longitudinal consistency

UMAP for the graph-based method is colored by pseudotime in Figure 3a and d, and Leiden clusters in Figure 3b-c. We display the trajectory of nodules that progress to lung cancer (Figure 3b and e) and those that remain as benign nodules (Figure 3c and f) in blue arrows. Most malignant nodules have arrows pointing toward higher pseudotime, while benign nodules remain mostly local and occasionally display back-and-forth movements. Visualizations of selected nodules are shown in Figure A1.

In Figure 4a and c, we show boxplots of Pearson correlations between inferred pseudotime and actual time for malignant and benign nodules. Nodules that later progress to malignancy exhibit significantly higher correlations across both methods. A similar pattern is observed in the Sankey diagrams (Figure 4b and d). A large portion of malignant nodules transition from lower pseudotime to higher pseudotime (green). On the other hand, benign cases predominantly maintain a stable pseudotime throughout the screening period (gray).

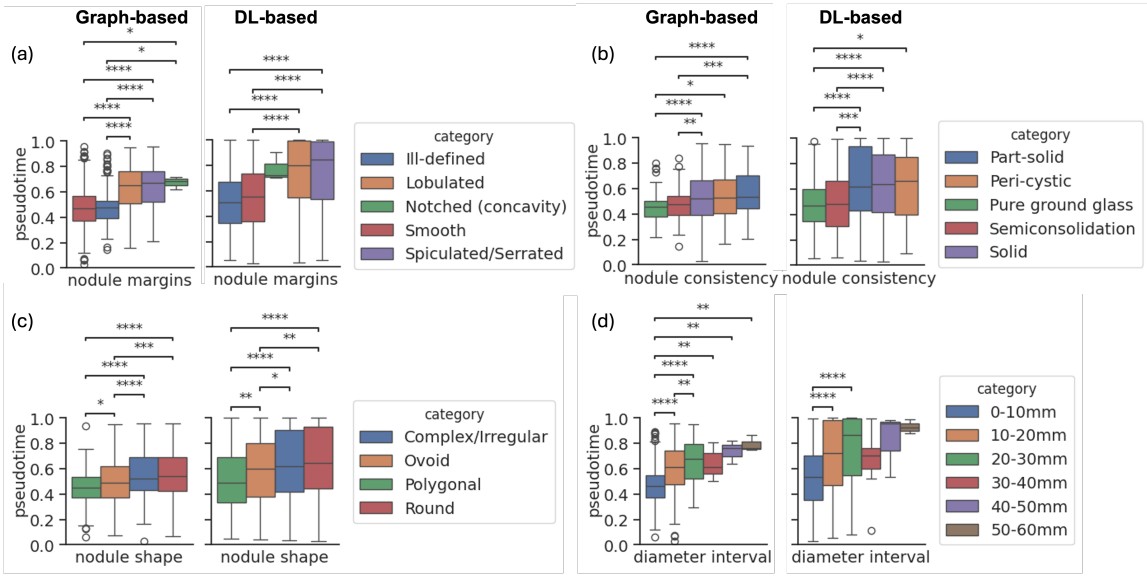

Figure 5: **Clinical Plausibility.** Boxplots illustrate differences in pseudotime across categories of nodule margin **(a)**, consistency **(b)**, shape **(c)**, and diameter interval **(d)**. Statistical significance was indicated by asterisks (*) based on the Mann–Whitney U test.

### 3.2. Pseudotime shows clinical relevance through semantic feature associations

To validate whether pseudotime preserves any clinical relevance, we present boxplots of pseudotime across different subtypes in semantic features, including nodule margin, consistency, shape, and longest axial diameter, as depicted in Figure 5. In both graph-based and DL methods (Figure 5a), nodules exhibiting suspicious margins, such as notched, spiculated, and lobulated, demonstrate significantly higher pseudotime than those with smooth margins. Furthermore, in Figure 5b, pseudotime for pure ground glass and semiconsolidation nodules is markedly lower than that for part-solid and solid nodules. In Figure 5c, pseudotime in both methods consistently increases from polygonal to ovoid, then to complex and round nodules. While we observe an increasing trend in pseudotime when the longest axial diameter of nodules increases, statistically significant differences are shown only between nodules in the 0–10mm range and those larger than 10mm. (Figure 5d).

### 3.3. Pseudotime and delta-pseudotime demonstrate distinct malignancy risk stratification and serve as independent predictors

Figure 6 shows Kaplan-Meier curves for three groups stratified by pseudotime and delta-pseudotime, respectively. All curves demonstrate statistically significant separation between the three groups, regardless of whether pseudotime or delta-pseudotime is used or which method is applied. Notably, the separation is more obvious when groups are stratified by delta-pseudotime. Pseudotime derived from the graph-based method shows greater differentiation than that from the DL-based method.

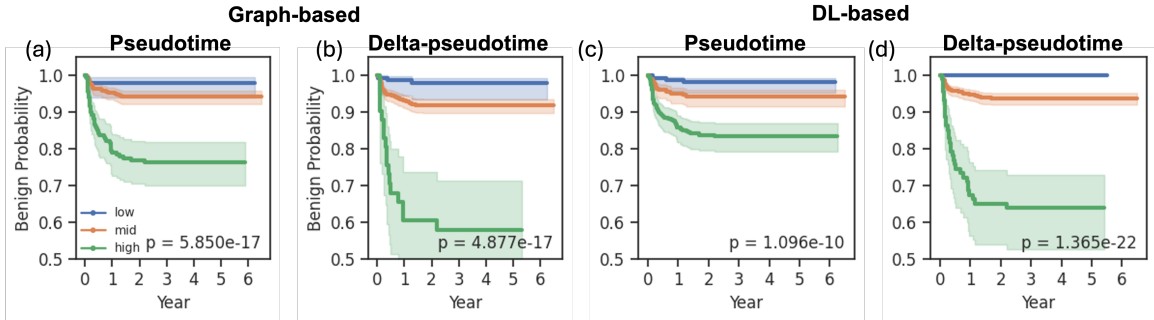

Figure 6: **Kaplan-Meier curves.** Kaplan-Meier curves show the proportion of benign cases across three groups defined by equal-width intervals of pseudotime and delta-pseudotime. P-values from the log-rank test are shown in the lower right.

As shown in Table 1, nodule margin, consistency, size, and changes in consistency and size reveal significant p-values in the Cox model. The graph-based method identifies both pseudotime (HR = 3.27, $p = 0.01$) and delta-pseudotime (HR = 18.54, $p < 0.01$) as statistically significant predictors. Nodules positioned later in the inferred trajectory and those with rapid change along the trajectory are more likely to be diagnosed as lung cancer earlier. In contrast, in the DL-based method, pseudotime is not statistically significant (HR = 2.31, $p = 0.11$), but delta-pseudotime remains significant (HR = 3.00, $p = 0.04$).

## 4. Discussion

In our study, we introduce pseudotime inference into a novel domain, medical imaging, to learn lesion progression from cross-sectional data. We utilize two distinct pseudotime inference approaches. The graph-based approach estimates pseudotime by modeling transitions on a data manifold derived from diffusion maps. In contrast, the DL-based approach employs unsupervised learning to capture dynamic patterns to reconstruct the features. We validate that pseudotime shows longitudinal consistency, with malignant nodules displaying a significantly higher correlation between pseudotime and actual timestamps. In addition, pseudotime demonstrates clinically plausible patterns, with nodules evolving from pure ground-glass to solid and from smooth to irregular margins as pseudotime increases.

In our Cox proportional hazards analysis of time to diagnosis, even after adjusting for established lung cancer biomarkers, pseudotime and delta-pseudotime still provide independent information in predicting lung cancer risk. Delta-pseudotime shows significance in both graph-based and DL-based approaches, indicating that changes along the predicted trajectory from cross-sectional data provide greater predictive value than absolute position. This finding is also supported by the Kaplan-Meier curves, which show that nodule groups stratified by delta-pseudotime exhibit more distinct malignancy profiles. This aligns with clinical intuition, as rapid changes in nodule characteristics often indicate aggressive cancer behavior. In addition, one explanation for pseudotime being significant in the graph-based but not the DL-based approach is that the graph-based trajectory was constructed using

Table 1: **Cox Proportional Hazards Model Result.** Pseudotime and delta pseudotime are jointly modeled with other semantic features. Reported values include hazard ratios (HR) with 95% confidence intervals (CI) and p-values.

| Feature | Subgroups | Graph-based | | DL-based | |
|---|---|---|---|---|---|
| | | HR [95% CI] | p-value | HR [95% CI] | p-value |
| Margins (Smooth) | Lobulated | 2.23 [1.47, 3.38] | **<0.01** | 5.47 [2.75, 10.85] | **<0.01** |
| | Ill-defined | 1.01 [0.66, 1.55] | 0.96 | 4.08 [1.48, 11.21] | **0.01** |
| | Notched | 0.72 [0.03, 16.16] | 0.84 | 0.00 [0.00, inf] | 1.00 |
| | Spiculated | 2.19 [1.45, 3.30] | **<0.01** | 4.75 [2.39, 9.43] | **<0.01** |
| Shape (Ovoid) | Polygonal | 0.75 [0.45, 1.27] | 0.29 | 0.00 [0.00, inf] | 1.00 |
| | Round | 1.23 [0.84, 1.78] | 0.29 | 1.34 [0.70, 2.56] | 0.38 |
| | Complex | 1.03 [0.75, 1.41] | 0.88 | 0.69 [0.39, 1.25] | 0.22 |
| Consistency (Pure Ground Glass) | Part-solid | 1.03 [0.65, 1.64] | 0.90 | 3.32 [0.65, 16.85] | 0.15 |
| | Semi-consolidation | 0.79 [0.44, 1.40] | 0.42 | 1.77 [0.24, 12.82] | 0.57 |
| | Peri-cystic | 1.32 [0.52, 3.33] | 0.56 | 3.61 [0.59, 21.91] | 0.16 |
| | Solid | 1.26 [0.89, 1.79] | 0.19 | 7.62 [1.53, 38.06] | **0.01** |
| Change in attenuation (Stable) | Decreased | 0.77 [0.23, 2.53] | 0.66 | 0.00 [0.00, inf] | 1.00 |
| | Increased (diffuse) | 1.82 [1.06, 3.12] | **0.03** | 1.94 [1.02, 3.72] | **0.04** |
| | Increased (focal) | 1.37 [0.76, 2.48] | 0.29 | 1.35 [0.62, 2.97] | 0.45 |
| Size | Diameter | 1.06 [1.03, 1.08] | **<0.01** | 1.09 [1.06, 1.12] | **<0.01** |
| Change in size (Stable) | Decreased | 0.88 [0.34, 2.25] | 0.79 | 0.96 [0.13, 7.22] | 0.97 |
| | Increased | 3.25 [2.30, 4.59] | **<0.01** | 6.64 [3.90, 11.31] | **<0.01** |
| Pseudotime | Pseudotime | 3.27 [1.32, 8.11] | **0.01** | 2.31 [0.82, 6.49] | 0.11 |
| | Delta-pseudotime | 18.54 [4.06, 84.62] | **<0.01** | 3.00 [1.07, 8.42] | **0.04** |

both the training and test sets. This likely facilitates more accurate placement of test-set nodules by capturing their global and local interactions with other training nodules.

Overall, the DL-based approach shows greater potential for pseudotime inference in large-scale, heterogeneous datasets than the graph-based method. While the test set must be incorporated with cross-sectional data for modeling in the graph-based method, the DL-based method allows inference on a new dataset without re-modeling, which is crucial for real-world deployment. Furthermore, the graph-based method requires manual selection of root nodules, which can be subjective and highly variable. On the other hand, the DL-based method automatically ranks nodules in the latent space. In addition, we observe that FM-CIB features are highly sensitive to the location of the nodule (Figure A2). In the UMAP visualization for graph-based pseudotime, there is a clearer distinction between central and

peripheral nodules. Figure A1b also confirms the issue. Even when the nodule's morphology changes, its peripheral position dominates the feature representation, resulting in minimal change in pseudotime. Although this problem persists in the DL-based method, it is less severe because the DL method learns to prioritize features relevant to ranking nodules, rather than being influenced by spatial bias. Future work will focus on strategies to prevent shortcut learning driven by peripheral position. Unlike the graph-based methods that can capture bifurcations and cycles in trajectories, the DL-based approach primarily delineates a single dominant trajectory, as the encoder maps embeddings to a one-dimensional vector representing nodule order. However, pulmonary nodules may diverge into distinct pathological subtypes. Modeling them with a single trajectory implicitly assumes a shared developmental pathway, which can obscure biologically meaningful distinctions in nodules. Although no branching structure was observed in the graph-based results in this study, this may reflect limitations of the feature embeddings used. Future work will explore DL frameworks that can be extended to model complex progression patterns. In addition, we plan to incorporate more cross-sectional data beyond screening scans, including nodules at different stages across various CT scan types, to gain a more comprehensive view of all types of nodules. Our pseudotime generation approach based on imaging features is broadly generalizable and can be applied to other cancer types to model lesion progression.

Compared to machine learning models, which perform binary classification on medical images, the pseudotime inference approach has several advantages across model training and clinical interpretation. First, forcing nodules into a dichotomous classification does not reflect the continuous nature of nodule states. For instance, within malignant nodules, some lesions may represent early-stage disease, such as adenocarcinoma in situ, whereas others may represent more advanced stages, such as invasive adenocarcinoma. In contrast, the pseudotime inference approach learns the intrinsic manifold of disease progression directly from nodules and assigns a continuous outcome. Second, most prior longitudinal modeling studies rely only on within-patient temporal data and exclude cross-sectional nodules. Such an approach overlooks that nodules sampled from cross-sectional data actually lie along a latent continuum of nodule development and may collectively be regarded as time-series representations. Although longitudinal data were not included in this study, we will integrate longitudinal data to further refine the inferred trajectory and improve the accuracy of pseudotime estimation. Finally, pseudotime values generated by our approach carry clearer biological meaning and are more intuitive for clinical interpretation. Classification models output probability scores that reflect the model's confidence in its malignancy prediction. In contrast, pseudotime represents the relative state of a nodule along a continuous morphological progression from benign to malignant. This framing provides a more clinically grounded description of the disease state rather than diagnostic uncertainty.

## 5. Conclusion

Pseudotime inference offers a dynamic framework for understanding lung nodule progression from cross-sectional CT imaging data. Both pseudotime and delta-pseudotime capture clinically relevant patterns and function as independent predictors of malignancy risk. These metrics can facilitate patient stratification, guide personalized follow-up strategies, and support earlier detection of lung cancer.

## Acknowledgments

This work is supported by NIH/National Cancer Institute U2C CA271898, U01 CA233370, and the Department of Veterans Affairs Merit Review I01BX005721.

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

## Appendix A. Example Trajectories

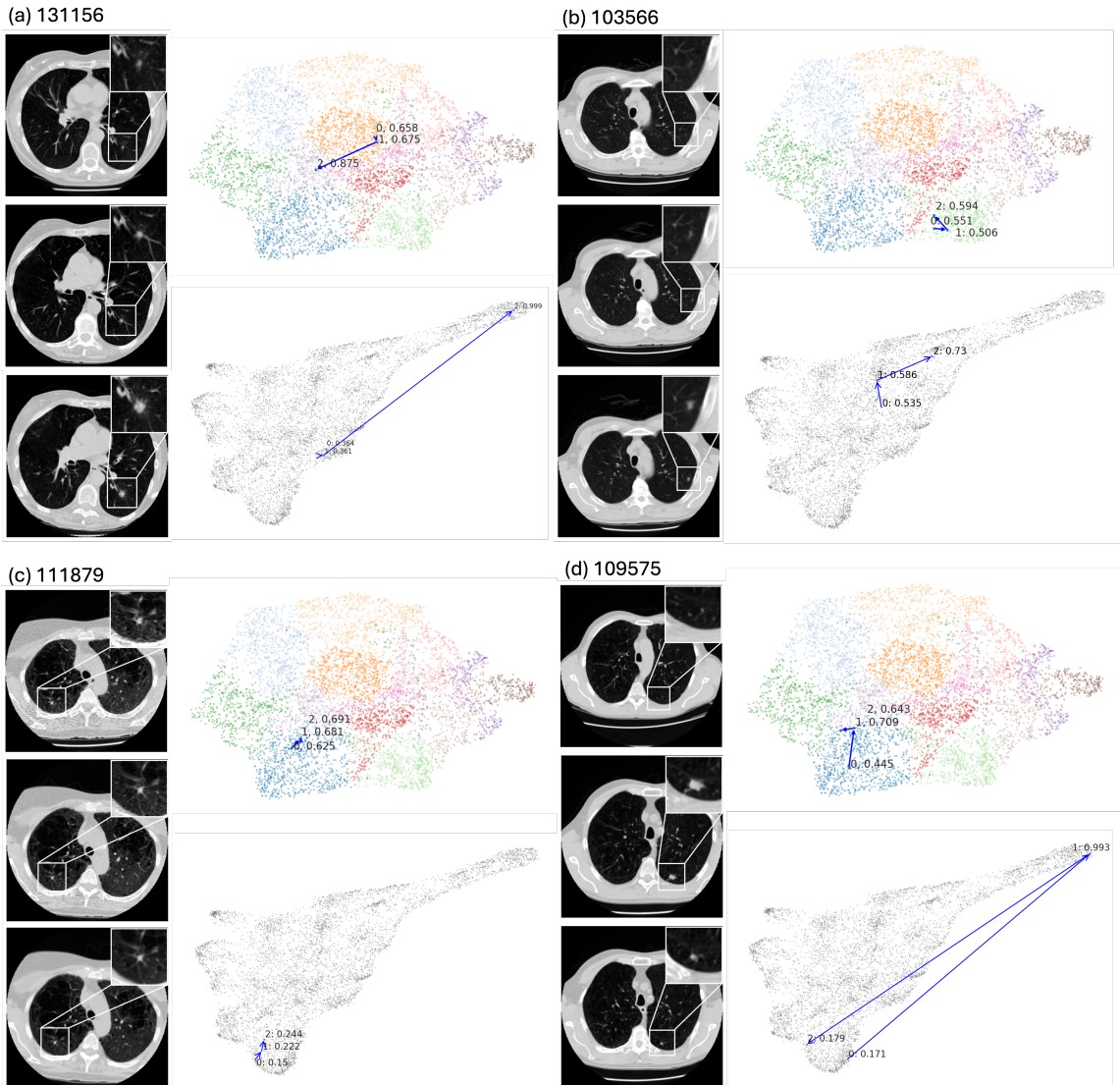

Figure A1: Examples. a–b show malignant nodules, and c–d show benign nodules. For each case, the CT slice containing the nodule is displayed with a magnified view in the top-right corner. Rows correspond to sequential timepoints from top to bottom, each with a one-year interval. The right side illustrates trajectories on UMAP from both methods, with pseudotime values annotated on the plots. The identifiers at the top correspond to the patient IDs (PID) from the NLST dataset.

Figure A1a-b depict two nodules that ultimately progress to lung cancer. Figure A1a illustrates a nodule that initially appears small, gradually enlarges, and eventually becomes significantly larger with spiculated margins. Pseudotime from both methods aligns with this

progression, remaining relatively stable in the first two timepoints and showing a drastic increase at the final timepoint. In Figure A1b, a gradually appearing nodule develops into a spiculated and irregular solid nodule. While the graph-based pseudotime remains the same, the DL-based pseudotime better reflects progression, showing an increasing trend. Figure A1c-d display two nodules that remain benign. The nodule in Figure A1c shows minimal change across three timepoints, and the pseudotime is stable. In Figure A1d, a large nodule starts to emerge in the second timepoint but subsequently decreases in size in the final year, which is also mirrored by an increase followed by a decrease in pseudotime.

## Appendix B. Batch Effect in Features

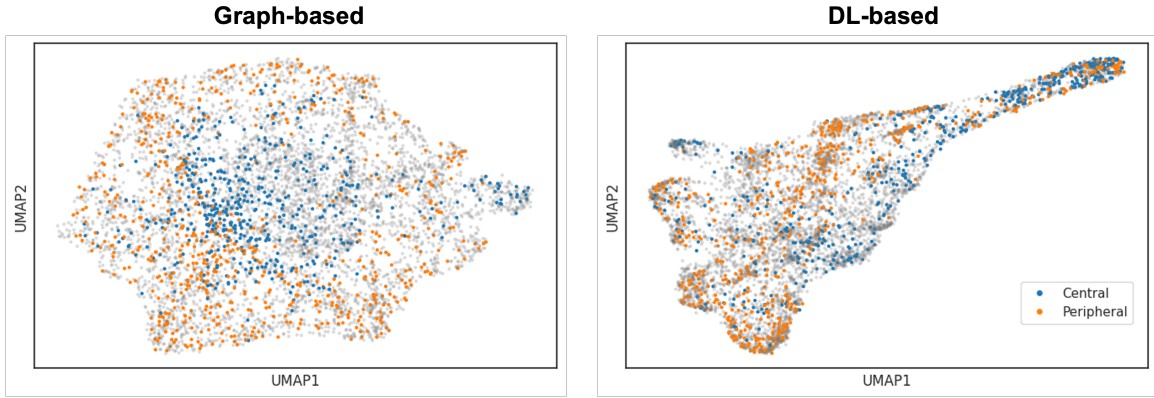

Figure A2: Effect of Axial Locations on Features. For a subset of nodules, axial location (central or peripheral) was annotated by radiologists and color-coded on the UMAP visualization.

