# OpenReview forum: "From Cross-Sectional CT to Dynamic Insights: Pseudotime-Based Modeling of Lung Nodule Progression"
_MIDL.io/2026/Conference — MIDL 2026 Poster_

### Official Review · Reviewer_G4s4 · 2026-01-09

**Confidence:** 5
**Preliminary Rating:** 4

**Summary:**

This article introduces the concept of pseudo-time inference to characterise long nodules in CT images. Data from the National Lung Screening Trial and Duke Lung Cancer Screening were filtered for sure cases with sparse nodules at several timepoints. A standard feature set was clustered and reduced using UMAP, and graph analysis was compared to a deep learning approach to model time trajectories. The results indicate that graph-based and deep-learning-based analyses behave differently and that deep-learning methods seem to identify malignant trajectories better than the graph-based method.

**Strengths:**

The work concerns an important and challenging problem of estimating temporal behaviour from temporally and sparsely sampled data. It reads well and introduces well-established methods from other fields into a new field. The paper analyses a large sample, and the results are encouraging.

**Weaknesses:**

The paper introduces so many methods that little space is left to properly explain the ideas. Therefore, it reads as an overview paper, and it is difficult to assess the importance of each element. For me, the complete lack of a summary of the features studied (section 2.2) makes it difficult. Further, the loss function (above Section 2.3.2) seems to be missing Lagrange parameters to model the influence of each term. I'm also lacking a discussion on the potential influence on the stochasticity of the UMAP (Section 3.1), and I found graphs like Figure 4 difficult to appreciate, as well as a discussion on the usefulness of the fitted lines in the scatter plots in Figure 5d.

**Detailed Comments:**

No detailed comments

**Justification Of The Preliminary Rating:**

The work on deducing temporal relations in sparsely sampled data is very important, and the tools appear strong. There is, nevertheless, so many details that it is impossible to fully grasp the influence of each step on the result.

**Questions To Address In The Rebuttal:**

See above.

---

> ### Author Response · Authors · 2026-01-25
>
> We appreciate the reviewer’s valuable comments on the paper's methodological details. We provide a point-by-point reply to each question raised.
>
> **Q1:** For me, the complete lack of a summary of the features studied (section 2.2) makes it difficult.
>
> **ANSWER:** We used a pretrained deep learning-based model, the Foundation Model for Cancer Imaging Biomarkers, to extract 4,096-dimensional features from nodule crops. Unlike traditional handcrafted features (e.g., radiomics features), which have explicit definitions, features derived from deep learning models are usually abstract and can be difficult to interpret. These features can encode complex visual information, such as texture, shape, and intensity patterns, and have been shown to have clinical and biological associations with cancer. We have expanded the explanation in Section 2.2 to clarify this (Page 4).
>
> **Q2:** Further, the loss function (above Section 2.3.2) seems to be missing Lagrange parameters to model the influence of each term.
>
> **ANSWER:** We have modified the loss function to include the Lagrange parameters, including alpha_z, alpha_h, beta, and gamma. We also specify in the text that we set alpha_z and alpha_h to 0.5, and beta and gamma to 1 (Page 6).
>
> **Q3:** I'm also lacking a discussion on the potential influence on the stochasticity of the UMAP (Section 3.1)
>
> **ANSWER:**  We would like to clarify that the primary outcome of this study is the pseudotime score, computed by the two pseudotime inference algorithms (Section 2.3). This process is independent of UMAP. UMAP is solely used as a post hoc visualization tool. We acknowledge that UMAP involves stochastic initialization, which may lead to variations in point layout. However, our findings do not rely on the specific geometric structure of the embedding. As shown in Figure 3a and d, we overlaid pseudotime values onto the embedding to facilitate visual interpretation. Combined with Figure 3b-c and e-f, we observed that malignant nodules tend to start at lower pseudotime and progress toward higher pseudotime values.
>
> In Section 2.4 Evaluation, we have updated the main text to explicitly clarify the end product of the algorithms, and UMAP is used to assess the concordance between inferred pseudotime and observed longitudinal evolution (Pages 6-7).
>
> **Q4:** I found graphs like Figure 4 difficult to appreciate, as well as a discussion on the usefulness of the fitted lines in the scatter plots in Figure 5d.
>
> **ANSWER:**  We replaced the line plots with Sankey diagrams in Figure 4 (Page 8). In the original line plots, we randomly selected 50 cases to show the changes in pseudotime for each. However, Sankey diagrams offer a more comprehensive view of pseudotime transitions across all cases. This improves clarity and removes overlap seen in the original figures. Sankey diagrams (Figure 4b, d) illustrate the pseudotime transitions for longitudinal scans of malignant and benign nodules, respectively. T0 indicates the final scan timepoint, which is when lung cancer is diagnosed for malignant nodules. Pseudotime values are grouped into 0.2 interval bins. In these diagrams, the thickness of flows reflects the relative number of nodules. We observed that a large portion of malignant nodules shift from lower to higher pseudotime (green). Conversely, benign cases mostly maintain a stable pseudotime throughout screening (gray).
>
> We also replaced the original scatter plot in Figure 5d with a box plot, which matches the format of the remaining subplots in Figure 5 (Page 9). We categorized the longest axial diameter into bins of 10mm. The Mann–Whitney U test was used to assess the statistical significance of differences in pseudotime. While we observe a positive trend in pseudotime with increasing nodular axial diameter, statistically significant differences are observed only between nodules in the 0–10mm range and those larger than 10mm.

---

### Official Review · Reviewer_RnTB · 2026-01-09

**Confidence:** 4
**Preliminary Rating:** 5
**Final Rating:** 5

**Summary:**

This study introduces a novel approach to model lung nodule progression using pseudotime inference, a method adapted from single-cell RNA sequencing, applied to cross-sectional CT imaging data. Two existing methods (a graph-based approach and a deep learning framework combining a variational autoencoder with neural ordinary differential equation) were adapted and evaluated for their ability to reconstruct dynamic trajectories of nodule progression. The algorithms were trained on cross-sectional CT-scans scans from over 6000 cases and 13000 nodules from the NLST and the DLCS datasets (preprocessing was not trivial). In the graph model, imaging features were extracted using a foundation model.
Results demonstrate that pseudotime correlates with actual progression time in malignant nodules and aligns with clinically relevant features, such as irregular margins and solid consistency. Pseudotime and delta-pseudotime emerged as independent predictors of malignancy.

**Strengths:**

The strengths of this paper are as follows:
-  first application of pseudotime inference to model the progression of lung nodules, thereby introducing a methodological framework from single cell analysis to the field of medical imaging.
- the study introduces a comparative analysis of two distinct approaches.
- the analysis is conducted on a large-scale dataset, which enhances the statistical significance and robustness of the findings.
- the results are highly compelling, and the discussion provides insightful interpretations.

**Weaknesses:**

The paper exhibits certain limitations, many of which are acknowledged by the authors:
- No method appears completely satisfactory (e.g. subjectivity for selection of the root nodule or bias related to nodule location for graph-based model, linear trajectories for DL approach).
- The absence of a comprehensive comparison with established clinical risk models.

**Detailed Comments:**

I would make the following remarks regarding the paper:
- In the legend of Figure 3, the term "Pseudotime" is missing the letter "d" and should be corrected to "Pseudotime."
- The term "delta-pseudotime" is introduced in the text prior to its formal definition on page 8. It would be advisable to define this term upon its first mention to avoid confusion.
- Clarification regarding the deep learning-based approach would be beneficial, particularly with respect to its tendency to delineate single linear trajectories, which may limit its capacity to capture more complex progression patterns.
- The availability of the code is not addressed in the paper. While it would be preferable for the code to be made accessible, providing additional details regarding the methodology and parameters would, at the very least, enhance the reproducibility of the approach.

**Justification Of Final Rating:**

I acknowledge that the authors’ revisions have improved the paper, which was already convincing. My remarks were taken into account in the revised version. I strongly recommend acceptance of this work.

**Justification Of The Preliminary Rating:**

I find this work solid and interesting. While the study does not introduce entirely new methods, it successfully adapts and applies robust approaches from single-cell RNA sequencing analysis to the domain of medical imaging. This (non-trivial) adaptation represents a novel and innovative framework for modeling lung nodule progression using cross-sectional CT data.
The paper demonstrates the relevance and effectiveness of this approach through a rigorous comparative analysis of graph-based and deep learning-based methods. The results are well-supported by comprehensive evaluations, including longitudinal consistency checks, clinical plausibility assessments, and statistical modeling, all of which underscore the potential of pseudotime inference as a valuable tool for dynamic lesion modeling. The clinical implications of the findings, particularly in risk stratification and personalized follow-up strategies, further enhance the paper's significance.

**Questions To Address In The Rebuttal:**

I do not have specific questions that require a response in the rebuttal. However, I would request that my remarks be taken into consideration.

---

> ### Author Response · Authors · 2026-01-25
>
> We thank the reviewer for the positive feedback and interest in the study. We provide a point-by-point response to each comment.
>
> **Q1:** No method appears completely satisfactory (e.g. subjectivity for selection of the root nodule or bias related to nodule location for graph-based model, linear trajectories for DL approach).
>
> **ANSWER:** We agree that both computational methods have trade-offs. Our primary goal in this study was to validate the concept by adapting established trajectory inference techniques from single-cell RNA-seq to radiological features. While we demonstrate that these methods yielded meaningful results, they may be better suited to gene expression data, and we will explore how to develop a model tailored to medical imaging features in the future. However, as mentioned in the Discussion section, we do think the DL approach has more potential due to better computational efficiency and automatic feature extraction, which can be adjusted to address biases.
>
> **Q2:** The absence of a comprehensive comparison with established clinical risk models.
>
> **ANSWER:** We acknowledge that we did not compare our results to existing clinical risk models. However, we addressed this conceptually with the multivariate Cox proportional hazards analysis. In our analysis, we adjusted for established lung cancer risk biomarkers, including nodule size, margin, shape, consistency, and changes in both size and consistency. Our results indicate that, even after accounting for these features, pseudotime and delta-pseudotime still provide independent prognostic information. We agree that comprehensive benchmarking will be essential for clinical translation. We plan to collect additional external data in the future, with labels, for more thorough comparison.
>
> **Q3:** In the legend of Figure 3, the term "Pseudotime" is missing the letter "d" and should be corrected to "Pseudotime."
>
> **ANSWER:** We have corrected the spelling in the Figure 3 caption (Page 7).
>
> **Q4:** The term "delta-pseudotime" is introduced in the text prior to its formal definition on page 8. It would be advisable to define this term upon its first mention to avoid confusion.
>
> **ANSWER:** We have defined delta-pseudotime in its first appearance in the text, Section 2.4 (Page 6).
>
> **Q5:** Clarification regarding the deep learning-based approach would be beneficial, particularly with respect to its tendency to delineate single linear trajectories, which may limit its capacity to capture more complex progression patterns.
>
> **ANSWER:** Pulmonary nodules may diverge into distinct pathological subtypes. However, in the DL-based method, the encoder maps embeddings onto a one-dimensional vector representing nodule order. Modeling them with a single trajectory implicitly assumes a shared developmental pathway, which can obscure biologically meaningful distinctions in nodules. Some graph-based methods, such as diffusion pseudotime, can capture bifurcations and cycles in the trajectory. Although we did not observe branching in the graph-based method in our study, this may be due to limitations in the feature embeddings we used, and we do not want to exclude the possibility of a more complex progression pattern in our future study. We have expanded our explanation in the Discussion section (Pages 11-12).
>
> **Q6:** The availability of the code is not addressed in the paper. While it would be preferable for the code to be made accessible, providing additional details regarding the methodology and parameters would, at the very least, enhance the reproducibility of the approach.
>
> **ANSWER:** The code of our implementation can be found here (https://github.com/luotingzhuang/Pseudotime4Nodules). Our implementation relies on publicly available open-source Python libraries. For the graph-based approach, we used the Scanpy package (version 1.11.5), following the standard protocol, which is available in the official documentation. For the deep learning approach, we utilized scTour implementation available on GitHub (https://github.com/LiQian-XC/sctour). We have updated the methods to provide more details of code implementation in Section 2.3.2 (Page 6). The hyperparameters were also described in the same section.

---

### Official Review · Reviewer_f9cV · 2026-01-10

**Confidence:** 4
**Preliminary Rating:** 4

**Summary:**

This paper presents the first application of pseudotime inference, a technique adopted from single-cell RNA sequencing, to reconstruction progression trajectories from cross-sectional CT images. Two methodologies are considered. One is a diffusion pseudotime estimated on a kNN graph constructed on image features extracted from a lesion foundation model. The other is a DL-based pseudotime estimation method known as scTour. Evaluation considered correlation between the pseudotime and real time, pseudotime between nodules with different features, and risk stratification ability of the estimated pseudotime.

**Strengths:**

While the methodology adopted is not new, this paper represents the first exploration of an interesting and under-explored area in medical imaging (progression modeling from cross sectional data)

As a first exploration, this evaluation study includes several interesting methods to evaluate the estiamted pseudotime and its relation to the underlying disease state.

**Weaknesses:**

Since this is essentially an estimated pseudotime from an image already observed, the clinical utility of the methodology is not quite clear. e.g., while the pseudotime is shown to be associated with the ability to differentiate benign vs. malicious nodules, one can ask what it benefit could be compared to directly differentiation using the image itself. The clinical use of the “progression modeling” is not quite clear.

The innovation of the work is limited to the application of two existing methods to a new area.

Fig 5d does seem to really show an evident trend as descried.

**Detailed Comments:**

See weakness section.

**Justification Of The Preliminary Rating:**

While applying existing methods without a clear clinical utility, this paper does represent a first analysis of pseudotime estimation for progression modeling from cross sectional medical images. It analyses overall are comprehensive and insightful. It could have good discussion value and may inspire further research on this topic in the community.

**Questions To Address In The Rebuttal:**

Clarification on the clinical utility of the pseudotime estimation could be helpful in better conveying the contribution of the work

---

> ### Author Response · Authors · 2026-01-25
>
> We thank the reviewer for the insightful questions regarding the clinical utility and novelty of the study. We provide a point-by-point response to each question raised.
>
> **Q1:** Since this is essentially an estimated pseudotime from an image already observed, the clinical utility of the methodology is not quite clear. e.g., while the pseudotime is shown to be associated with the ability to differentiate benign vs. malicious nodules, one can ask what it benefit could be compared to directly differentiation using the image itself. The clinical use of the “progression modeling” is not quite clear.
>
> **ANSWER:** We identify several advantages of pseudotime inference over direct binary differentiations:
> - Granularity in modeling the disease continuum: A binary classifier assigns labels 0 or 1 to benign and malignant cases, collapsing biologically heterogeneous states into discrete labels. However, the nodule’s state can be continuous. For malignant nodules, some can be at an early stage, such as adenocarcinoma in situ, while others can be at a later stage, such as invasive adenocarcinoma. A binary classifier treats these distinct biological states identically as 1, losing the prognostic nuance of how far the cancer has progressed. On the other hand, the pseudotime approach directly infers the intrinsic manifold of disease progression from images.
> - Better clinical interpretability: In terms of clinical interpretation, the risk score output from the binary classification model represents the likelihood of the label, i.e., the probability that the nodule is malignant. However, the pseudotime generated reflects the state along the morphological progression from benign to malignant. This provides a specific description of the current biological state rather than just diagnostic uncertainty.
> We have updated the Discussion section to address this (Page 12).
>
> **Q2:** The innovation of the work is limited to the application of two existing methods to a new area.
>
> **ANSWER:** The innovation of this work lies in translating widely used single-cell RNA-seq methods to medical imaging. This paper is one of the first to demonstrate this, particularly in lung cancer. However, we introduce a very different perspective or paradigm from that currently used in machine learning applications in medical imaging.  As stated in the introduction, most work on longitudinal modeling still emphasizes within-patient time-series data. However, it overlooks the fact that nodules collected from cross-sectional data actually lie on a latent continuum of nodule development. In addition, as mentioned in Q1, such an approach would provide greater granularity in reflecting the nodule’s current state and better clinical interpretability than the binary classification approach used in most studies now. This work serves as the feasibility study for this domain. We will continue to improve the methodological innovation in pseudotime application for nodule progression in the future.
>
> **Q3:** Fig 5d does seem to really show an evident trend as descried.
>
> **ANSWER:** We interpret the reviewer’s comment as questioning whether Fig. 5d clearly shows the described trend. We have replaced the original scatter plot in Figure 5d with a box plot (Page 9). This can better show the distinction in pseudotime across different nodule size groups and align with the format of the remaining subplots in Figure 5. We categorized the longest axial diameter into bins of 10mm. The Mann–Whitney U test was used to assess the statistical significance of differences in pseudotime. While we observe an increasing trend in median pseudotime with increasing nodular axial diameter, statistically significant differences are observed only between nodules in the 0–10mm range and those larger than 10 mm.

---

### Author Rebuttal · Authors · 2026-01-25

**Rebuttal:**

We sincerely thank all reviewers for constructive feedback. We have attached the revised manuscript here. All changes have been highlighted in red.

We have addressed the main points below:

**Innovation & Clinical Utility:** We clarify the clinical utility of pseudotime as a continuous biological manifold that captures disease granularity beyond simple binary classification. It also offers more biologically meaningful interpretability than traditional models. This work represents a novel paradigm shift in medical imaging to inferring latent development from cross-sectional data.

**Improved visualization:** To strengthen the visualization in the manuscript, we have replaced the original plots with Sankey diagrams (Fig. 4) and boxplots (Fig. 5d) to demonstrate progression trends more clearly.

**Reproducibility:** We provided a public code repository to ensure reproducibility: https://github.com/luotingzhuang/Pseudotime4Nodules.

The point-by-point response is available in the official comments. Thank you for your time and consideration. We look forward to further discussions.

**Supporting Material:**

/attachment/2f7fa57dfcf7f02b97d4ab51a8c5faa56af72f2a.pdf

---

### Comment · Area_Chair_f8hZ · 2026-01-30

Dear Reviewers,

Please ensure your final score reflects your current assessment. You can update it by clicking “Edit” → “Official Review” and providing the Final Rating.

Thank you.

---

### Meta-Review · Area_Chair_f8hZ · 2026-02-07

**Recommendation:** Accept (Oral)
**Confidence:** 5

**Metareview:**

The reviewers were broadly positive and agreed that applying pseudotime inference to CT imaging is a novel and interesting direction, even if the methods themselves are adapted from other fields.
The main concerns were about limited methodological novelty, unclear clinical utility, missing methodological details, and a few unclear figures. The rebuttal addressed these points clearly: the authors expanded the clinical motivation, clarified methodological components, corrected figure issues, and improved visualizations.
Considering the overall reviewer agreement and the satisfactory rebuttal, I recommend acceptance of this paper.

---

### Decision · Program_Chairs · 2026-02-13

Accept (Poster)